# In the Seeking of Association between Air Pollutant and COVID-19 Confirmed Cases Using Deep Learning

**DOI:** 10.3390/ijerph19116373

**Published:** 2022-05-24

**Authors:** Yu-Tse Tsan, Endah Kristiani, Po-Yu Liu, Wei-Min Chu, Chao-Tung Yang

**Affiliations:** 1Department of Emergency Medicine, Taichung Veterans General Hospital, Taichung City 407204, Taiwan; janyuhjer@gmail.com; 2School of Medicine, Chung Shan Medical University, Taichung City 40201, Taiwan; williamchu0110@gmail.com; 3Division of Occupational Medicine, Department of Emergency Medicine, Taichung Veterans General Hospital, Taichung City 407204, Taiwan; 4Department of Computer Science, Tunghai University, No. 1727, Sec. 4, Taiwan Boulevard, Taichung City 407224, Taiwan; endahkristi@thu.edu.tw; 5Department of Informatics, Krida Wacana Christian University, Jakarta 11470, Indonesia; 6Division of Infection, Department of Internal Medicine, Taichung Veterans General Hospital, Taichung City 407204, Taiwan; pyliu@vghtc.gov.tw; 7Department of Family Medicine, Taichung Veterans General Hospital, Taichung City 407204, Taiwan; 8School of Medicine, National Yang Ming Chiao Tung University, Taipei City 11221, Taiwan; 9Department of Post–Baccalaureate Medicine, College of Medicine, National Chung Hsing University, Taichung City 40227, Taiwan; 10Institute of Health Policy and Management, National Taiwan University, Taipei City 10617, Taiwan; 11Research Center for Smart Sustainable Circular Economy, Tunghai University, No. 1727, Sec. 4, Taiwan Boulevard, Taichung City 407224, Taiwan

**Keywords:** COVID-19, AQI, air pollutant, correlation analysis, deep learning, LSTM, lag times

## Abstract

The COVID-19 pandemic raises awareness of how the fatal spreading of infectious disease impacts economic, political, and cultural sectors, which causes social implications. Across the world, strategies aimed at quickly recognizing risk factors have also helped shape public health guidelines and direct resources; however, they are challenging to analyze and predict since those events still happen. This paper intends to invesitgate the association between air pollutants and COVID-19 confirmed cases using Deep Learning. We used Delhi, India, for daily confirmed cases and air pollutant data for the dataset. We used LSTM deep learning for training the combination of COVID-19 Confirmed Case and AQI parameters over the four different lag times of 1, 3, 7, and 14 days. The finding indicates that CO is the most excellent model compared with the others, having on average, 13 RMSE values. This was followed by pressure at 15, PM_2.5_ at 20, NO_2_ at 20, and O_3_ at 22 error rates.

## 1. Introduction

Although we remember and contemplate that during 26 January–3 October 2020, more than 300,000 people died in the United States, with two thirds of those deaths directly associated to COVID-19 [1], we might also assess what the newest science says about the pandemic. We know that those who live in places with severe levels of air pollution will face several hazards concerning their respiratory health throughout this outbreak. Currently, new research focuses on the correlations between air pollution and severe COVID-19 sickness, emphasizing the crucial need for everyone to breathe clean air. Research published in December 2020 attempted to assess the extent to which COVID-19 mortality is due to long-term exposure to fine particle pollution [2]. Using a combination of epidemiological data, satellite data, and other monitoring data worldwide, the researchers concluded that chronic air pollution might be responsible for 15% of COVID-19 fatalities globally [2]. The experts also distinguished air pollution generated by fossil fuels and pollution that is induced by other human activities. In the United States, fossil fuel-related air pollution is responsible for 15% of COVID-19 mortality, indicating that fossil fuel related air pollution contributes considerably to overall U.S. air quality [2].

Several researchers used machine learning to evaluate the association and prediction cases when investigating the relationship between two parameters [3,4]. Machine learning is commonly used in drug development research to anticipate chemical characteristics and help identify active molecules. Perez and Bajorath [5] provide a new method to discover links between target proteins that incorporate internal model information from chemical activity predictions. Feature importance correlation analysis is proven to discover comparable drug binding features based on a large-scale investigation that compared machine learning models for more than 200 proteins. Surprisingly, the research also finds functional links between proteins that are not dependent on active chemicals or binding properties. Malki et al. [6] also provide several regressor machine learning algorithms to extract the association between different parameters and the COVID-19 distribution rate. By extracting the association between the number of confirmed cases and the climatic variables in specified places, the machine learning methods used in this study assess the influence of weather variables such as temperature and humidity on COVID-19 transmission. As a result of this finding, they may deduce that temperature and humidity are critical factors in determining COVID-19 death rates. Furthermore, it has been shown that the greater the temperature, the lower the number of infection cases. To collectively predict the correlated time series, Wan et al. [7] suggested CTS-LSTM, a unique variation of LSTM. Experiments are carried out on two types of real-world datasets: civil aviation passenger demand data and air quality data. In addition, when compared with state-of-the-art baselines, CTS-LSTM achieves at least 9.0%, 16.5%, and 21.3% reduced RMSE, MAE, and MAPE. An LSTM recurrent neural network-based technique for predicting the load of non-residential customers utilizing various correlated sequence information is provided in [8]. The suggested methodology is evaluated on a real data set containing energy consumption data from 48 non-residential users in China. The results of the experiments suggest that this strategy can effectively use numerous sequence information and successfully capture the relationships between different sequences.

Therefore, based on the previous research on the potential correlation of two or more different resources conducted by some experts, we examined the relationship between air pollutants and COVID-19 using deep learning. The objectives of this paper are listed as follows:To correlate COVID-19 Confirmed Case with AQI parameters.To train the integration of COVID-19 Confirmed Case and AQI parameters in four different lag times, 1, 3, 7, and 14 days, using long short-term memory (LSTM) deep learning.To evaluate and compare the RMSE values for the trained models.

The contribution of this paper might leverage the research of correlation and prediction analysis of air pollutants and COVID-19 using different approaches, such as lag times and LSTM methods combined with correlation analysis.

## 2. Background Review and Related Work

Cui et al. [9] discovered that the residents of a severely polluted area of China were more likely to die of SARS than residents in a less polluted area. Kan et al. [10] discovered that increases in particulate matter air pollution enhanced the probability of dying from the disease during the 2003 SARS pandemic. Numerous viruses, including adenovirus and influenza virus, have been proven to be transmitted by air particles. Zhao et al. [11] concluded that particulate matter was probably a factor in the propagation of the 2015 avian influenza. According to Chen et al. [12], air pollution can hasten the spread of respiratory diseases.

### 2.1. Research on Association of Air Pollutant and COVID-19

Researchers conducted the study related to the association of air pollution and COVID-19 in various countries. We examined their works to enrich our knowledge of this topic, as follows.

Zhu et al. [13] investigated the association between ambient air pollution and coronavirus infection. Between 23 January 2020 and 29 February 2020, in China, daily confirmed cases, air pollution concentrations, and climatic data were collected in 120 cities. They used a generalized additive model to examine the relationships between six air pollutants (PM_2.5_, PM_10_, SO_2_, CO, NO_2_, and O_3_) and verified instances of COVID-19.

Gupta et al. [14] estimated the increased risk of coronavirus disease (COVID-19), caused by severe acute respiratory syndrome coronavirus 2, by establishing a link between the mortality rate of infected individuals and air pollution, specifically Particulate Matters (PM) with aerodynamic diameters of 10 m and 2.5 m. Nine Asian cities’ data are studied using statistical techniques such as analysis of variance and regression modeling.

Lolli et al. [15] quantified the relationship between COVID-19 transmission and meteorological and air quality indices in two significant urban regions in Northern Italy, Milan, and Florence, as well as the autonomous province of Trento. Milan, the capital of the Lombardy region, is often regarded as the heart of Italy’s HIV epidemic.

Bashir et al. [16] investigated the relationship between COVID-19 and climatic indicators in New York City, United States of America. They analyzed secondary public data from the New York City Department of Health and the National Weather Service in the United States of America. The average temperature, lowest temperature, maximum temperature, rainfall, average humidity, wind speed, and air quality are all covered in the research. The Kendall and Spearman rank correlation tests were used to analyze the data.

Suhaimi et al. [17] investigated the relationships between air quality, climatic variables, and COVID-19 cases in Kuala Lumpur, Malaysia. The Department of Environment Malaysia provided air pollutants and meteorological data from 2018–2020, whereas the Ministry of Health Malaysia provided daily new COVID-19 case data in 2020.

Mehmood et al. [18] used geospatial tools to analyze the relationship between COVID-19 cases, air pollution, meteorological, and socioeconomic characteristics in three provincial capital cities and the federal capital city of Pakistan.

Hoang and Tran [19] investigated the temporal association in seven metropolitan centers and nine regions across Korea using the generalized additive model. The findings indicate a substantial nonlinear relationship between daily temperature and verified COVID-19 cases.

Travaglio et al. [20] matched current SARS-CoV-2 cases and fatalities from public databases to regional and subregional air pollution data collected across England.

In Singapore, Lorenzo et al. [21] examined the relationship between core air pollutant concentrations, climatic factors, and daily verified COVID-19 case numbers. We collected data on air pollutant concentrations (particulate matter [PM_2.5_, PM_10_], ozone [O_3_], carbon monoxide [CO], nitrogen dioxide [NO_2_], sulphur dioxide [SO_2_], pollutant standards index [PSI]), and climatic variables (rainfall, humidity, and temperature). Table 1 summarizes recent studies on the association between air pollutants and COVID-19. 

Recent important research findings in 2022 also indicated that there is a correlation between air pollutants and COVID-19 cases [22,23,24,25,26].

### 2.2. Research on Prediction of Air Pollutant and COVID-19 Using Deep Learning

Aragão et al. [27] examined climate factors as extra features in a data-driven multivariate prediction model to predict the number of COVID-19 deaths in Brazilian states and significant cities in the short future. The basic premise is that by including these climatic characteristics as inputs to data-driven model training, the prediction performance increases when compared with single-input models. For both the multivariate and univariate models, the training adopted a Stacked LSTM as the network architecture. Using the mean fitting error, average forecast error, and the profile of the cumulative deaths for the forecast as evaluation criteria, the tests revealed that the best multivariate model is more skillful than the best standard data-driven univariate model we found. These findings suggest that by using additional important variables as input for a multivariate method, the quality of prediction models may be improved even more.

Al-Qaness et al. [28] presented an upgraded version of the adaptive neuro-fuzzy inference system (ANFIS) for forecasting the air quality index in Wuhan City, China. The PSOSMA is a hybrid optimization approach that uses a novel modified meta-heuristics (MH) algorithm, and a slime mold algorithm (SMA), which is enhanced by employing the particle swarm optimizer to increase ANFIS performance (PSO). The proposed PSOSMA-ANFIS was trained using three years of air quality index time series data and then used to forecast fine particulate matter (PM_2.5_), sulfur dioxide (SO_2_), carbon dioxide (CO_2_), and nitrogen dioxide (NO_2_) for a year. The suggested PSOSMA was also compared with various MH algorithms used to train ANFIS. The results discovered that the improved ANFIS incorporating PSOSMA outperformed the other methods.

Zhou et al. [29] discussed the COVID-19 forecasting, using the relevance of government initiatives in their suggested model, the Interpretable Temporal Attention Network (ITANet). Long short-term memory (LSTM) for temporal feature extraction and multi-head attention for the long-term dependency caption are used in the proposed model, which has an encoder–decoder architecture. The ITANet outperforms other models when it comes to anticipating COVID-19 new confirmed cases.

Saravanan et al. [30] described the impact of lockdown measures on air quality and rainwater accumulation in major cities. With respect to varying time length and climatic variables, the effects of COVID-19 on the environment during lockdown conditions were compared with those without lockdown conditions. During the lockdown, the concentrations of particulate pollution in Chennai, Bangalore, Delhi, and Melbourne were measured. The findings of this research indicate the effects of government actions and give a detailed perspective of the death rate in relation to air quality decrease.

Xu, et al. [31] created three deep learning models in their study to forecast the number of COVID-19 cases for Brazil, India, and Russia, including CNN, LSTM, and CNN-LSTM. The LSTM model, among the models constructed in this study, has the best forecasting performance, which indicates an improvement in prediction accuracy over certain current models.

Fu, et al. [32] used experimental public data sets from the Johns Hopkins University Center for Systems Science and Engineering (JHU CSSE), the Air Quality Open Data Platform, the China Meteorological Data Network, and the WorldPop website. The Dual-link Bi-GRU Network predicts the epidemic scenario, and the Gauss–Newton iteration method quantifies the relationship between epidemic spread and other feature parameters. Among the selected characteristic elements, the study discovered that population density had the most positive link with pandemic spread, followed by the number of landing planes.

Mumtaz, et al. [33] suggested an indoor air quality monitoring and prediction system based on the newest Internet of Things (IoT) sensors and machine learning capabilities, which can assess a variety of indoor pollutants. An IoT node including numerous sensors for eight pollutants, including NH_3_, CO, NO_2_, CH_4_, CO_2_, PM_2.5_, as well as the ambient temperature and air humidity, has been designed for this purpose. With an accuracy of 99.37%, precision of 99%, recall of 98%, and F1-score of 99%, this model has showed promise in forecasting air pollutants’ concentrations as well as overall air quality.

### 2.3. LSTM Network

In nonlinear sequence prediction issues, the LSTM network is a recurrent neural network (RNN) design that can learn order dependence. They have a habit of memorizing things for a long period. The memory cell, which substitutes classic neurons’ hidden layers, is at the foundation of the LSTM network [34]. The LSTM networks, similarly to other RNNs, feature recurrent cells, but instead of a single NN gate, the recurring cell has an interactive input gate, output gate, and forget gate [35]. The cell remembers values for arbitrary time intervals, and these three gates control the flow of information into and out of the cell. Based on the past state, accessible memory, and current input, this structure guarantees that the LSTM can recognize which cells are stimulated and compressed. The LSTM networks were created to solve the problem of disappearing gradients that might occur when training traditional RNNs. As there may be unexpected delays between critical occurrences in a time series, LSTM networks are ideally suited for categorizing, processing, and generating predictions based on time series data. In many cases, LSTM has an advantage over RNNs, hidden Markov models, and other sequence learning approaches due to its relative insensitivity to gap length; therefore, we selected LSTM as the model to predict the integration of COVID-19 and air pollutant data.

## 3. Materials and Methods

In this section, we presented the materials and methods, including the dataset used in this paper, the research workflows, and the LSTM training method.

### 3.1. Dataset

The dataset was extracted from different resources, as follows.

India’s COVID-19 dataset per state.AQI in India per State.These two datasets were then integrated based on Delhi as the case study.From the data integration of AQI and COVID-19, we obtained 609 records from the period of 2 March 2020 to 31 October 2021.

Based on these resources, we obtained the parameters, as described in Table 2.

### 3.2. Research Workflows

First, we collected the dataset from two resources. The first resource is from the Indian government’s COVID-19 dataset per state, from which, Delhi was selected. The second resource was the AQI parameters based on city/state (again, we selected Delhi). Then, we integrated these two resources based on per day values. After that, we separated the lag time between air pollution and COVID-19 confirmed cases for 1, 3, 7, and 14 day lag times. In this case, the majority approach of lag time selections rely on trials to determine the best time-lags, which may not always be sufficient in real-world circumstances [36,37,38,39]. These approaches, on the other hand, are mostly based on trial-and-error scenarios, which necessitates the training of various models multiple times in order to identify the best among them. Next, we train the dataset using LSTM and use the models [40,41,42]. Figure 1 shows the workflows of this research.

### 3.3. Data Preprocessing

In machine learning, data preparation is a critical step that helps improve data quality and facilitates the extraction of relevant insights from the data [43,44]. After we integrated COVID-19 confirmed cases and air pollutants, we completed the data preprocessing. The data preprocessing was conducted as follows.

For handling the missing values, we marked all NA values with 0.To make sure that the calculations are fine, we ensured that all data were floats.To complete the standardization of all values, we normalized all features.Then, we converted our time-series data to a supervised learning problem.For the model’s training requirements, we split the dataset into training and test sets.Next, we paired the input and outputs from the data sequence.For LSTM model, we needed to reshape the input into 3D (samples, timesteps, features).

### 3.4. LSTM Training Modelling

The design network for the training model is illustrated in Figure 2 and Figure 3, as follows.

The model was compiled with the setting: model.compile(loss = ‘mae’, optimizer = ‘adam’)EarlyStopping(monitor = “val_loss”, patience = 20, verbose = 1, mode = “auto”)history = model.fit(train_X, train_y, epochs = 200, batch_size = 72, validation_data = (test_X, test_y), validation_split = 0.33, verbose = 1, shuffle = False, callbacks = [callback])

The training was based on MAE Loss, with an Adam Optimizer. We implemented the EarlyStopping method to avoid overfitting. The fit network was set in 200 epochs, and a 72 batch size.

To make a prediction, the process is as follows.

Feed the model into test data.Invert scaling for forecast data.Invert scaling for actual data.Calculate RMSE.

## 4. Results

Based on the designed experiments, we have 28 models for comparison. The results are as follows.

### 4.1. Matrix Correlations

Based on the matrix correlation in Figure 4, it can be seen that there are 3 parameters that have a strong positive correlation, which are pressure, NO_2_, and PM_2.5_ at 0.53, 0.45, and 0.42, respectively.

### 4.2. Model Training Results

The purpose of the learning algorithm is to find a decent match between an overfit and an underfit model. A good fit is defined as a training and validation loss that declines to the point of stability with a slight difference between the two final loss values. The model’s loss is usually always smaller than the validation dataset on the training dataset. It implies that a divergence between the training and validation loss learning curves should be expected. If the training loss plot drops to the point of stability, the plot of learning curves reveals a satisfactory match. The validation loss plot reaches a point of stability, with a tiny gap between it and the training loss, as shown in Figure 5, Figure 6, Figure 7, Figure 8, Figure 9, Figure 10, Figure 11, Figure 12, Figure 13, Figure 14, Figure 15, Figure 16, Figure 17 and Figure 18. Figure 5, Figure 7, Figure 9, Figure 11, Figure 13, Figure 15 and Figure 17 illustrate that all the learning curves were a good fit; therefore, when plotted, the prediction looks to be substantially closer to the test set, as shown in Figure 6, Figure 8, Figure 10, Figure 12, Figure 14, Figure 16 and Figure 18.

COVID-19 Confirmed Cases and all AQI Parameters

**Figure 5 ijerph-19-06373-f005:**
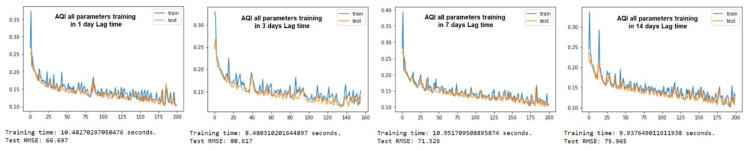
Training and test of all AQI parameters during the 1, 3, 7, and 14 day lag times.

**Figure 6 ijerph-19-06373-f006:**
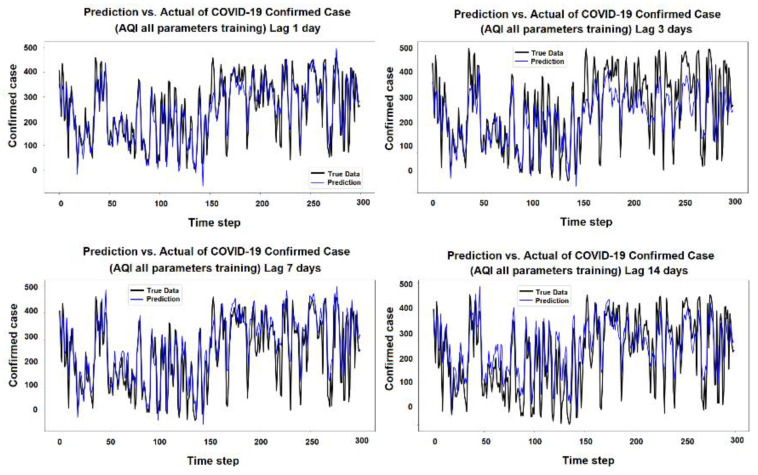
Prediction vs. actual results of AQI parameters during the 1, 3, 7, and 14 day lag times.

COVID-19 Confirmed Cases and PM_2.5_ Parameters

**Figure 7 ijerph-19-06373-f007:**
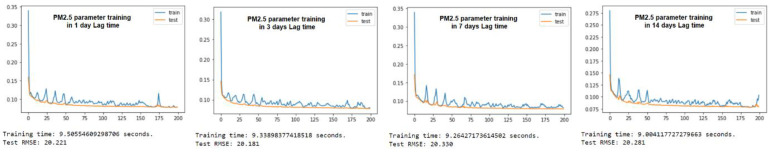
Training and test for PM_2.5_ during the 1, 3, 7, and 14 day lag times.

**Figure 8 ijerph-19-06373-f008:**
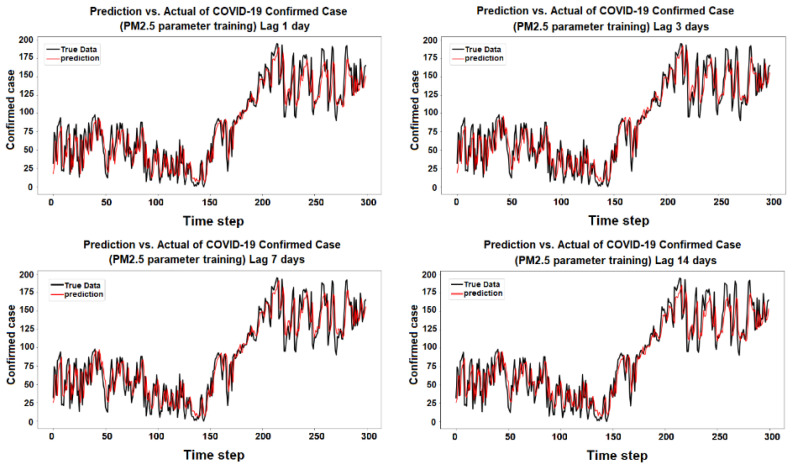
Prediction vs. actual results of PM_2.5_ during the 1, 3, 7, and 14 day lag times.

COVID-19 Confirmed Cases and NO_2_ Parameters

**Figure 9 ijerph-19-06373-f009:**
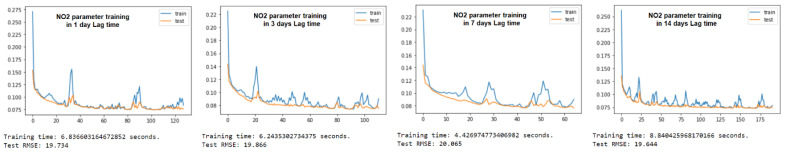
Training and test for NO_2_ during the 1, 3, 7, and 14 day lag times.

**Figure 10 ijerph-19-06373-f010:**
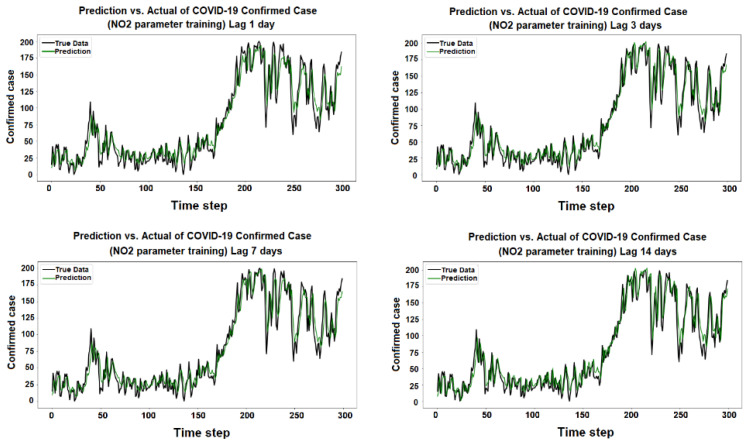
Prediction vs. actual results of NO_2_ during the 1, 3, 7, and 14 day lag times.

COVID-19 Confirmed Cases and Pressure Parameters

**Figure 11 ijerph-19-06373-f011:**
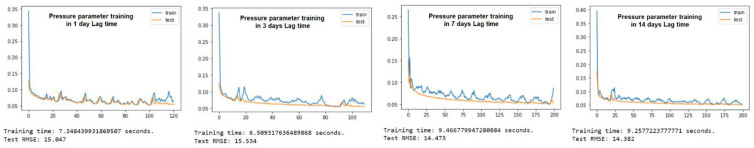
Training and test for pressure during the 1, 3, 7, and 14 day lag times.

**Figure 12 ijerph-19-06373-f012:**
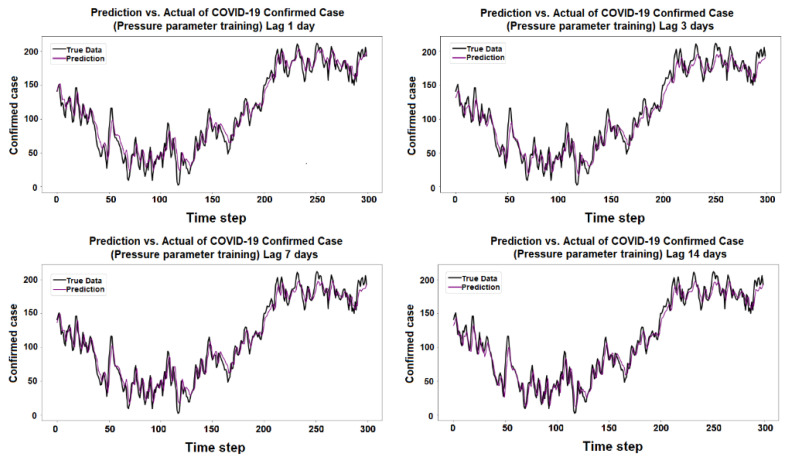
Prediction vs. actual results of pressure during the 1, 3, 7, and 14 day lag times.

COVID-19 Confirmed Cases and O_3_ Parameters

**Figure 13 ijerph-19-06373-f013:**
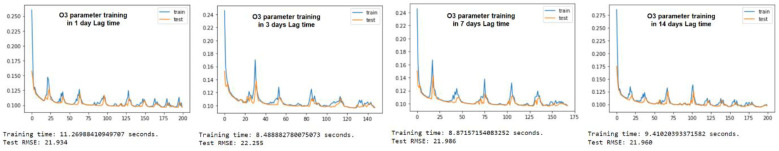
Training and test for O_3_ during the 1, 3, 7, and 14 day lag times.

**Figure 14 ijerph-19-06373-f014:**
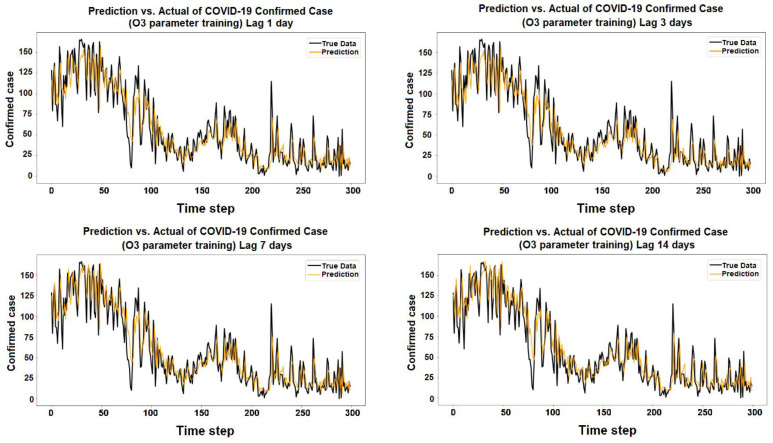
Prediction vs. actual results of O_3_ during the 1, 3, 7, and 14 day lag times.

COVID-19 Confirmed Cases and CO Parameters

**Figure 15 ijerph-19-06373-f015:**
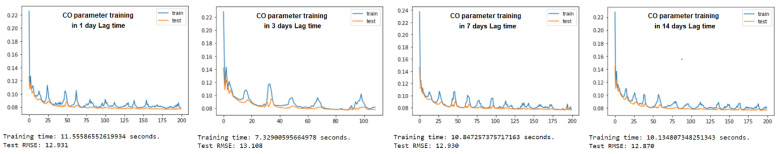
Training and test for CO during the 1, 3, 7, and 14 day lag times.

**Figure 16 ijerph-19-06373-f016:**
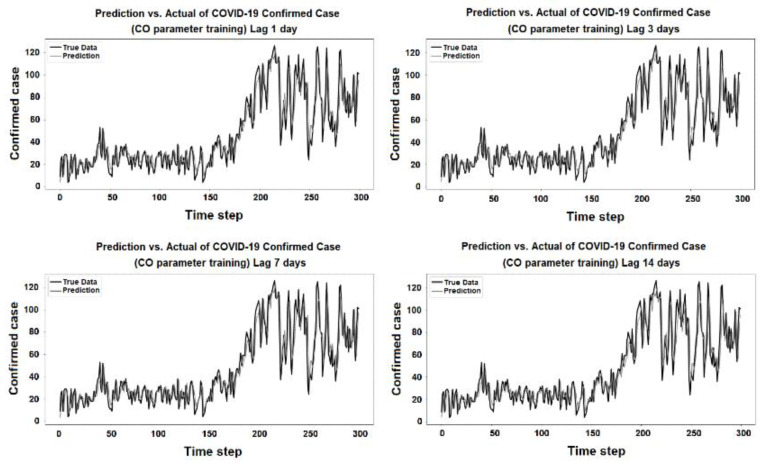
Prediction vs. actual results for CO during the 1, 3, 7, and 14 day lag times.

COVID-19 Confirmed Cases and Humidity Parameters

**Figure 17 ijerph-19-06373-f017:**
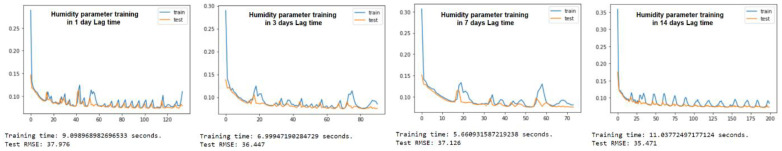
Training and test for humidity during the 1, 3, 7, and 14 day lag times.

**Figure 18 ijerph-19-06373-f018:**
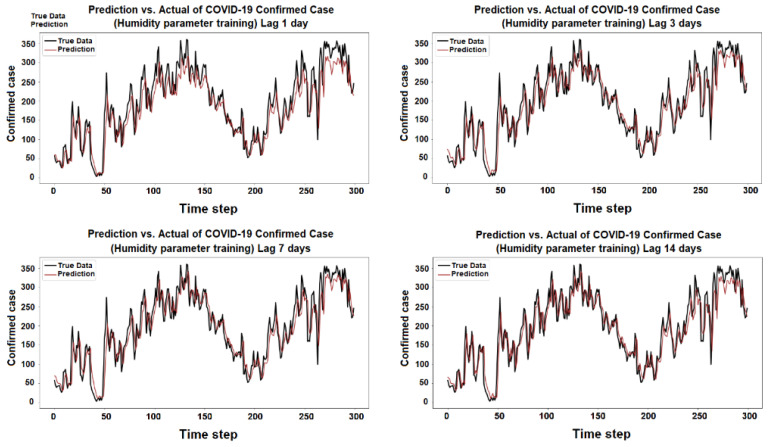
Prediction vs. actual results for Humidity during the 1, 3, 7, and 14 day lag times.

### 4.3. RMSE and Variance Model Comparison

The RMSE comparison graph in Figure 19 illustrates the comparison of LSTM models of COVID-19 confirmed cases and air pollutant parameters. The dataset was divided into four lag times: 1, 3, 7, and 14 days.

The first model contains all air pollutant parameter training, which uses 12 parameters, PM_2.5_, PM_10_, WIND_SPEED, WIND_GUST, O_3_, CO, Humidity, Pressure, Dew, NO_2_, Precipitation, and WIND_DIREC. The results show that the RMSE values are the highest compared with the other models, at 66.6967 for the 1 day lag time, 80.6172 for the 3 day lag time, 80.6172 for the 7 day lag time, and 75.9652 for the 14 day lag time, respectively.The second model uses PM_2.5_ parameter training. It can be seen from the results that the RMSE values reach 20.2212 for the 1 day lag time, 20.1812 for the 3 day lag time, 20.1812 for the 7 day lag time, and 20.2813 for the 14 day lag time, respectively.The third model uses NO_2_ parameter training. It can be seen from the results that the RMSE values reach 19.7336 for the 1 day lag time, 19.8656 for the 3 day lag time, 19.8656 for the 7 day lag time, and 19.6443 for the 14 day lag time, respectively.The fourth model uses pressure parameter training. It can be seen from the results that the RMSE values reach 15.0469 for the 1 day lag time, 15.5343 for the 3 day lag time, 15.5343 for the 7 day lag time, and 14.382 for the 14 day lag time, respectively.The fifth model uses O_3_ parameter training. It can be seen from the results that the RMSE values reach 21.9336 for the 1 day lag time, 22.2551 for the 3 day lag time, 22.2551 for the 7 day lag time, and 21.9604 for the 14 day lag time, respectively.The sixth model is the most excellent model compared to the others, it uses CO parameter training. It can be seen from the results that the RMSE values reach 12.931 for the 1 day lag time, 13.1081 for the 3 day lag time, 13.1081 for the 7 day lag time, and 12.8699 for the 14 day lag time, respectively.Finally, the seventh model uses humidity parameter training. It can be seen from the results that the RMSE values reach 37.9753 for the 1 day lag time, 36.4467 for the 3 day lag time, 36.4467 for the 7 day lag time, and 35.4713 for the 14 day lag time, respectively.

The explained variance is a metric for determining how much variability exists in a machine learning model’s predictions. To put it another way, it is the difference between the expected and forecasted values. Understanding how much information we can lose by reconciling the dataset is a crucial subject. At least 60% of the variation in a machine learning model must be explained. The goal is to have a value that is low. From the Graph Variance Score Comparison in Figure 20, it can be seen that the fourth model (pressure parameter training) has excellent variance scores at more than 0.9, whereas the first model has the worst variance scores. NO_2_, PM_2.5_, CO, and humidity models have a score of more than 0.8, which is also acceptable.

## 5. Conclusions and Future Work

This study investigates the correlation of COVID-19 confirmed cases with AQI parameters using deep learning. The dataset was divided into four lag times, 1, 3, 7, and 14 days. From the lag times experiments, it can be found that one day lag time has an excellent RMSE. The deep learning models were good using the association of COVID-19 and air pollutants in the 1 day lag time scenario. We also performed the correlation in the matrix correlation coefficient, and the results show that the orders are pressure, NO_2_, PM_2.5_, PM_10_, CO, and O_3_, followed by humidity; however, these orders were different when we trained using deep learning. Seven models have experimented with deep learning LSTM, and COVID-19 confirmed cases with all air pollutant parameters, PM_2.5_, NO_2_, pressure, O_3_, CO, and humidity. From the training models, we found that CO is the most excellent model compared with the others, having on average, 13 RMSE values. CO is followed by pressure at 15, PM_2.5_ at 20, NO_2_ at 20, O_3_ at 22, humidity at 37, and finally, all air pollutant parameters at 76. As a result of the finding, we assume that CO, Pressure, NO_2_, and PM_2.5_ have a significant role in COVID-19 confirmed rates. In the future, more machine learning algorithms can be conducted to compare these results. Moreover, other data resources, such as people mobility, social media, and other countries’ data, might be analyzed in deep experiments.

## Figures and Tables

**Figure 1 ijerph-19-06373-f001:**
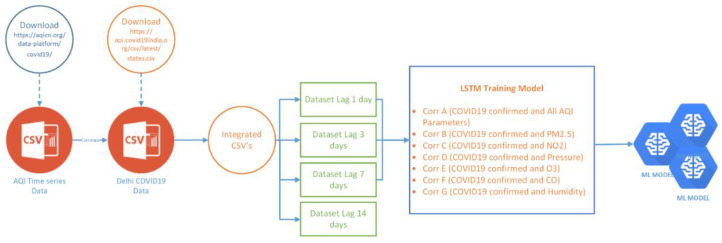
Research Workflow.

**Figure 2 ijerph-19-06373-f002:**
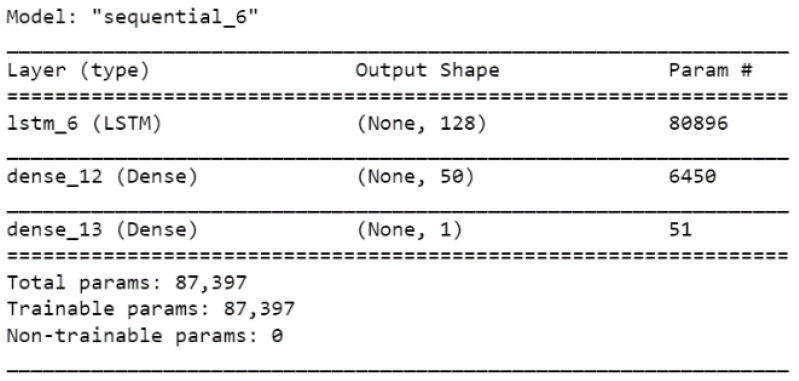
LSTM layers model.

**Figure 3 ijerph-19-06373-f003:**
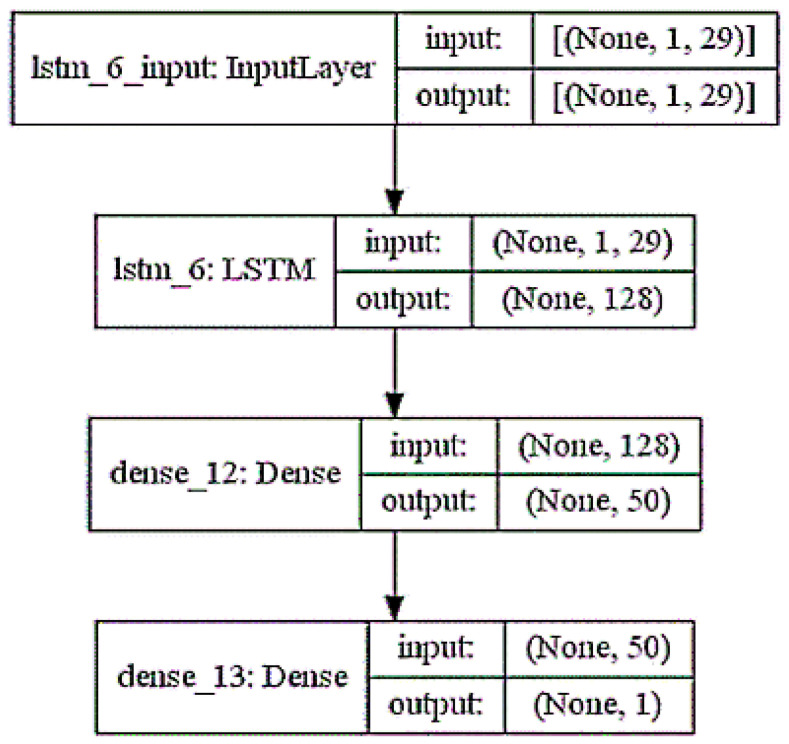
LSTM model plot.

**Figure 4 ijerph-19-06373-f004:**
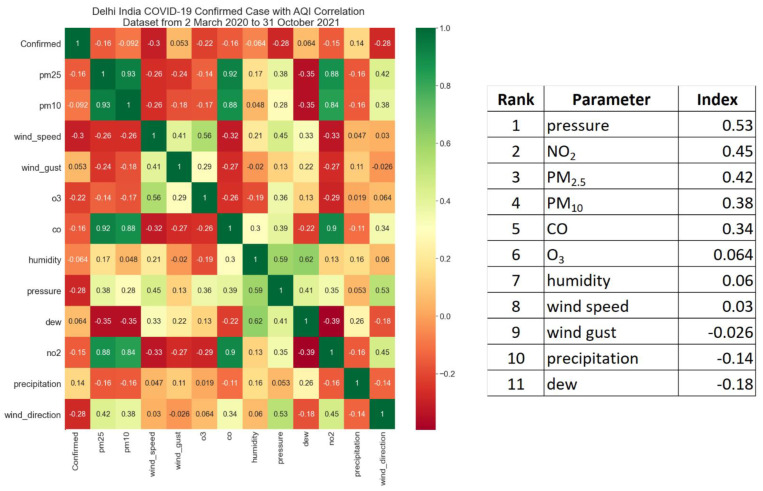
Matrix correlation of air pollutants and COVID-19.

**Figure 19 ijerph-19-06373-f019:**
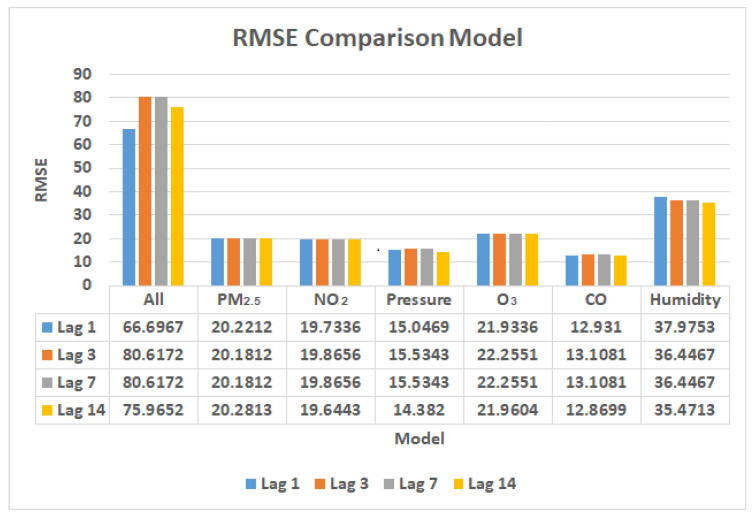
RMSE comparison models for the 1, 3, 7, and 14 day lag times.

**Figure 20 ijerph-19-06373-f020:**
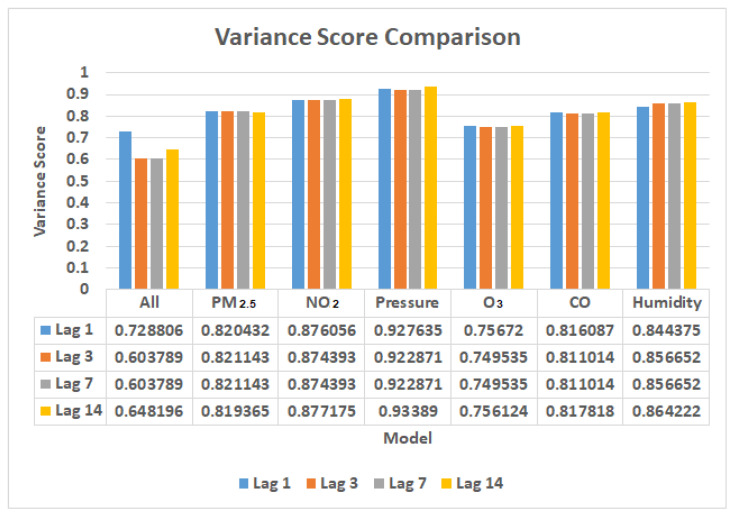
Variance score comparison in 1, 3, 7, and 14 lag times.

**Table 1 ijerph-19-06373-t001:** Recent research on Association of Air Pollutant and COVID-19.

Author (Year)	Objective	Location	Finding
Zhu et al. (2020)	Investigate the link between ambient air pollution and the new coronavirus infection.	One-hundred and twenty cities in China	PM_2.5_, PM_10_, NO_2_, and O_3_ levels were significantly higher in the previous two weeks in areas with newly confirmed COVID-19 cases.
Gupta et al. (2021)	Calculate the elevated incidence of coronavirus disease (COVID-19) induced by severe acute respiratory syndrome coronavirus 2 by demonstrating a correlation between the death rate of infected individuals and air pollution, especially Particulate Matters (PM).	Nine cities in Asia, Delhi India, Nagpur India, Kanpur India, Islamabad Pakistan, Lahore Pakistan, Jakarta Indonesia, Tianjin China, Guilin China, Hebei China	There is a positive association between a region’s degree of air pollution and the mortality associated with COVID-19, demonstrating that air pollution is a significant and hidden factor exacerbating the worldwide burden of COVID-19-related mortality.
Lolli et al. (2020)	The correlation between meteorological and air quality indicators and COVID-19 transmission was quantified.	Northern Italy, Milan, and Florence	Although elements such as temperature and humidity are inversely connected with viral transmission, air pollution (PM_2.5_) is positively correlated (to a lesser degree).
Bashir et al. (2020)	The connection between COVID-19 and climatic factors was analyzed.	New York City, USA	The COVID-19 pandemic was substantially related with average temperature, lowest temperature, and air quality.
Suhaimi et al. (2020)	Establish connections between air quality, climatic variables, and COVID-19 instances.	Kuala Lumpur, Malaysia	Spearman’s correlation analysis revealed a positive connection between COVID-19 cases and PM_10_ (r = 0.131, *p* < 0.001), PM_2.5_ (r = 0.151, *p* < 0.001), SO_2_ (r = 0.091, *p* = 0.003), NO_2_ (r = 0.228, *p* < 0.001), CO (r = 0.269, *p* = 0.001), and relative humidity (RH) (r = 0.106, *p* = 0.001).
Mehmood et al. (2021)	Using geospatial approaches to examine the connection between COVID-19 instances, air pollution, meteorological, and socioeconomic characteristics.	Three out of four provinces of Pakistan (Punjab, Sindh, Khyber Pakhtunkhwa)	The findings reveal that daily COVID-19 is positively linked with PM_2.5_ and other meteorological variables, implying that climate has a significant role in determining the COVID-19 incidence rate in Pakistan.
Hoang and Tran	The generalized additive model was used to evaluate the temporal connection between ambient air pollution, weather, and COVID-19 infection.	Seven metropolitan cities and nine provinces across Korea	Daily temperature had a substantial nonlinear relationship with verified COVID-19 cases.
Travaglio et al. (2021)	Evaluated recent SARS-CoV-2 cases and fatalities from public databases to regional and subregional air pollution data collected at several locations.	England	There is a positive correlation between COVID-19 mortality and infectivity and air pollution concentrations, notably nitrogen oxides.
Lorenzo et al. (2021)	Determine the relationship between core air pollutant concentrations, climatic factors, and daily verified COVID-19 cases.	Singapore	There is a statistically significant positive correlation between NO_2_, PSI, PM_2.5_, and temperature and COVID-19 case numbers.
Mandalapu et al. (2022)	The link between air pollution and COVID-19 severity has been studied at the regional and metropolitan levels, but it is uncertain if this link holds true at the neighborhood level.	Los Angeles County, California	Eighteen of the twenty-three significant comparisons for the COVID-19 weekly death rate confirmed that NO_2_ levels were higher in neighborhoods with higher COVID-19 weekly death rates. Similarly, 12 of the 19 comparisons confirmed the same relationship with CO levels, as 14 of the 23 comparisons confirmed the same relationship with ozone levels, and 6 of the 6 comparisons confirmed the same relationship with PM_10_.
Sidell et al. (2022)	To examine at both long-term and short-term air pollution exposure, as well as COVID-19 occurrence, from 1 March 2020 to 28 February 2021.	Southern California	In all case peaks before February 2021, long-term PM_2.5_ and NO_2_ exposures were linked to an elevated probability of COVID-19 occurrence. Short-term exposures to PM_2.5_ and NO_2_ were also linked. Air pollution may have a role in raising the likelihood of COVID-19 infection.
Luo et al. (2022)	This study assessed the relationship between population movement and air quality in 332 Chinese cities from January to March (2019–2021), and the influence of three city factors (pollution level, city scale, and lockdown status) in this impact.	Three-hundred and thirty-two Chinese cities	Lower migration was linked to lower pollution levels (other than O_3_). Susceptibility to pollution changes is more probable as NO_2_ decreases and O_3_ increases, whereas insusceptibility to pollution is more likely for CO and SO_2_, and in cities with low migration. Cities with less air pollution and dense populations may benefit the most from lowering PM_10_ and PM_2.5_. Those with rigorous traffic limits have higher links with population movement and air pollution than cities without limitations. The impacts of inter-city migration (ICM) and within city migration (WCM) on air pollution were found to be minor when city characteristics were considered.
Abdullah et al. (2022)	The connection between the Air Pollution Index (API) and COVID-19 infections is the objective of this research.	Malaysia	Each area has a positive connection between API and COVID-19: North 0.4% (R^2^ = 0.004), Central 2.1% (R^2^ = 0.021), South 0.04% (R^2^ = 0.0004), East 1.6% (R^2^ = 0.016), Sarawak 0.2% (R^2^ = 0.002), whereas Sabah has a negative correlation of 4.3% (R^2^ = 0.043).
Huang et al. (2022)	Data on air pollution and verified COVID-19 cases were collected from five severely affected cities in three South American nations. COVID-19’s spread was measured using daily real-time population regeneration (Rt). The influence of environmental contaminants on the pandemic was investigated using two commonly used models: generalized additive models (GAM) and multiple linear regression.	South America	(1) In all five locations, Rt, which potentially represents COVID-19 dissemination, exhibited a progressive drop. (2) Rt had a substantial effect on PM_10_ and SO_2_ in all of the locations studied. These two contaminants should be better monitored by regulators. (3) In cities with varying levels of air pollution, the link between air pollution and the spread of COVID-19 varied. The results indicate that there is a significant relationship between air pollution and COVID-19 infection.

**Table 2 ijerph-19-06373-t002:** Dataset parameters.

No.	Parameter	Unit	Name
1	COVID-19 Confirmed	Daily cases	The number of persons infected by COVID-19 in Delhi, India
2	PM_2.5_	µg/m^3^	Fine aerosol
3	PM_10_	µg/m^3^	Aerosol
4	WIND_SPEED	m/s	Wind speed
5	WIND_GUST	Knots	A sudden burst in wind speed
6	O_3_	Part per billion (ppb)	Ozone
7	CO	Part per million (ppm)	Carbon monoxide
8	Humidity	(g/kg)	Water vapor per kilogram of air
9	Pressure	Atmosphere (atm)	Air pressure
10	Dew	Celsius	Temperature
11	NO_2_	Part per billion (ppb)	Nitrogen dioxide
12	Precipitation	Millimeter (mm)	Water vapor
13	WIND_DIREC	Degree	Wind direction hourly

## Data Availability

The data are available in a publicly accessible repository that does not issue DOIs. Publicly available datasets were analyzed in this study. This data can be found here: https://api.COVID-19india.org/csv/latest/states.csv and https://aqicn.org/data-platform/COVID-19/ accessed on 1 January 2022.

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
