# Peer review of "In the Seeking of Association between Air Pollutant and COVID-19 Confirmed Cases Using Deep Learning"

_ijerph, 2022, doi:10.3390/ijerph19116373_

Round 1
Reviewer 1 Report
The article "In the Seeking of Association between Air Pollutant and COVID-19 Confirmed Cases Using Deep Learning" presented by Yu-Tse Tsan and co-authors can be interesting and useful to the community studying Covid-19 and the parameters which accompany its spread. Depite many negative concerns, this article can be accepted after major revision, in my opinion.
Unfortunately, authors started with a weak introduction to their work, literature research is not sufficient.
In the Abstract: (The finding indicates found) you used two successive verbs together. The sentence must be corrected.
Within the all text, all chemical formula must be written correctly: NO2 (not NO2), O3, CO2, SO2, NH3, CH4, … etc.
In the text, you must decide whether to write “COVID19” or “COVID-19”
Lines 198 and 199: they should not contain a direct link. You can add the links to different part, sup mat, or to the data availability part at the end of the article.
Similarly, Page 6, line 208: the link must be removed, you can add it as supplementary materials or to data. Otherwise as a comment at the end but not in the text.
Introduction, line 38: authors mentioned the number 300,000 dead in USA without any references and in which period of time. This must be corrected.
Line 43: “Research published in December 2020” which article? You must cite the article
Lines 45 to 47: references must be added here
Line 47 : (15% of COVID-19 fatalities globally) citation required
Line 48-50: citation required
Reference “1” is not accepted, please cite a considerable article. “American Lung Association” blog cannot be used as a reference. Moreover, the data they used is not cited and they have no references.
Line 48-50: The following paragraph must be all removed: “According to the findings, … and only 1% in New Zealand”
In citation, references 24 and 25 were used before reference 2, this must be corrected. References must be numbered in correct order.
Line 138: (examine at using) rewrite it in a correct way.
Page 15, line 341: (we might assume) remove “might” - you can use something like “we assume” or “we found”
Talking about old research should in in past form: Aragão et al. Examined - Al-Qaness et al. Presented - Zhou et al. discussed - Saravanan et al. Described - Fu, et al. used - … etc
Reference 7, 9, 13, 17, 23: contains “…” in the names of authors, you must write all names of authors and coauthors, you can check the citation rules of MDPI.
Figure 19 and 20 are not well aligned, all figures should be have the same position in the page.
Overall, more articles should be cited, English writing should be verified.
Author Response
Dear editors and reviewers,
Thank you for giving the constructive comments to enhance the manuscript entitled “In the Seeking of Association between Air Pollutant and COVID-19 Confirmed Cases Using Deep Learning” (Manuscript ID ijerph-1728575). It is our pleasure to have the second evaluation for publication. The authors have considered all reviewers’ suggestions, and the manuscript provides higher readability and completeness in the reversion. The authors would like to provide the revised manuscript and the comment reply, also we will English proofreading to the MDPI English service. We hope that the current version is qualified to be considered for publication in the International Journal of Environmental Research and Public Health MDPI.
Sincerely Yours,
Chao-Tung Yang Ph.D.

Reviewer 2 Report
Air pollution is a hot issue in the field of environment and epidemiology which has received more and more public concerns. And analyzing the relationship between air pollution and epidemics is crucial to human health. So, I believe that this problem selected by the authors of the present draft will attract lots of readers to follow. In addition, the structure of the draft is relatively complete. However, there are still a lot of problems in the draft that requires careful revision by the authors. I express my concerns as follows.
- Introduction section: although the research dynamic of the relation between air pollutants and COVID-19 was described by the authors, the unsolved issues in terms of using the deep learning method to detect the correlation between air pollutants and COVID-19 cases were not indicated well. So, the significance of the present study was not stated well. I strongly suggest that the authors strengthen this part accordingly.
- I suggest the author strengthen the data processing part to facilitate readers to have a more thorough understanding.
- LSTM Training Modelling section: the entire name of LSTM should be supplied the first time. The detailed information on the LSTM model such as math equations is strongly desired.
- I noticed that Lag: 1, 3, 7, and 14 days were selected as cohort studies in the current study. But the scientific support for this Lag selection scheme was not found in the current version. The authors should clarify why this scheme can be used in the draft.
- The conclusion part needs to be strengthened, the discussion was too simple, and the explanation especially the mechanism of the COVID-19 is expected.
- I strongly suggest that the authors invite a native speaker to polish the draft to assuring the correctness of the grammar.
Author Response

(The authors gave the same response as above.)

Round 2
Reviewer 1 Report
Dear authors,
Thank you for your response, and the modifications you did.
In your response, you reported that you removed the following paragraph, but I remind you that you forgot to delete it from the text:
"According to the findings, air pollution caused 27% 50 of COVID-19 fatalities in China, 18% in the United States, 15% in Mexico, 14% in the 51 United Kingdom, and 6% in Israel".
As I mentioned in my previous report, you did not give any scientific proof on this information. Thus it must be removed from the text.
Best regards,
Author Response
Dear editors and reviewers,
Thank you for giving the constructive comments to enhance the manuscript entitled “In the Seeking of Association between Air Pollutant and COVID-19 Confirmed Cases Using Deep Learning” (Manuscript ID ijerph-1728575-R2). It is our pleasure to have the second evaluation for publication. The authors have considered all reviewers’ suggestions, and the manuscript provides higher readability and completeness in the reversion. The authors would like to provide the revised manuscript and the comment reply, also we will English proofreading to the MDPI English service. We hope that the current version is qualified to be considered for publication in the International Journal of Environmental Research and Public Health MDPI.
Sincerely Yours,
Chao-Tung Yang Ph.D.

Reviewer 2 Report
The paper has been carefully revised and the quality was improved significantly. However, there still exists some issues before publication.
I express my concerns below.
1.Table 1 Dataset parameters. Figure 19 RMSE comparison models in 1, 2, 7, and 14 lag times. Some unints should be subscript. For example: m3,NO2.
Please keep consistency across entire manuscript.
2.Table 2. Recent research on Association of Air Pollutant and COVID-19. I suggest that authors search the related studies carefully, some important references in terms of the relation between air pollutants and COVID-2019 were ignored. Therefore, I still ask authors to add them accordingly.
3.2.3. Research on Prediction of Air Pollutant and COVID-19 Using Deep Learning. Although authors stated some deep learning contents, the unsolved issues especially using deep learning to conduct COVID-2019 related studies was not described. The reasons for selecting LSTM were also not clear. Please clarify.
4.4. Results. Authors listed the results' figures , but discriptions and analysis for some figures were inadqueate. For example, section 4.2. Please supplement.
5.Discussions for results were still not enough. Please strengthen.
Author Response

(The authors gave the same response as above.)
